# The New Nematicide Cyclobutrifluram Targets the Mitochondrial Succinate Dehydrogenase Complex in *Caenorhabditis elegans*

**DOI:** 10.3390/jdb11040039

**Published:** 2023-10-19

**Authors:** Fariba Heydari, David Rodriguez-Crespo, Chantal Wicky

**Affiliations:** Department of Biology, University of Fribourg, Chemin du Musée 10, 1700 Fribourg, Switzerland; fariba_heidari70@yahoo.com (F.H.); david.rodriguezcrespo@unifr.ch (D.R.-C.)

**Keywords:** *Caenorhabditis elegans*, cyclobutrifluram, cytochrome P450s, detoxifying proteins, RNA-seq, SDH (succinate dehydrogenase)

## Abstract

Today, agriculture around the world is challenged by parasitic nematode infections. Plant-parasitic nematodes (PPNs) can cause significant damage and crop loss and are a threat to food security. For a long time, the management of PPN infection has relied on nematicides that impact not only parasitic nematodes but also other organisms. More recently, new nematicides have been developed that appear to specifically target PPN. Cyclobutrifluram belongs to this new category of nematicides. Using the nematode *Caenorhabditis elegans* as a model organism, we show here that cyclobutrifluram strongly impacts the survival and fertility rates of the worm by decreasing the number of germ cells. Furthermore, using a genetic approach, we demonstrate that cyclobutrifluram functions by inhibiting the mitochondrial succinate dehydrogenase (SDH) complex. Transcriptomic analysis revealed a strong response to cyclobutrifluram exposure. Among the deregulated genes, we found genes coding for detoxifying proteins, such as cytochrome P450s and UDP-glucuronosyl transferases (UGTs). Overall, our study contributes to the understanding of the molecular mode of action of cyclobutrifluram, to the finding of new approaches against nematicide resistance, and to the discovery of novel nematicides. Furthermore, this study confirms that *C. elegans* is a suitable model organism to study the mode of action of nematicides.

## 1. Introduction

Nematodes are the most abundant animals on earth, and they represent a total biomass of approximately 0.3 gigatons [1]. Among nematodes species, plant-parasitic nematodes (PPNs), such as the root-knot nematode *Meloidogyne incognita*, strongly affect crop production and represent a significant threat to food security [1,2,3]. To counteract their devastating effects, fumigant nematicides, such as methyl bromide and chloropicrin, are widely used in agriculture. These nematicides have a large spectrum of activity on weeds, fungi, and nematodes, but they also have strong negative impacts on human health and the environment [4,5]. They have been gradually replaced by other nematicides, such as organophosphates or carbamates, whose mode of action relies on the inhibition of acetylcholine esterase. However, the impact of those is also not limited to nematodes; they are highly toxic to humans and insects [6,7]. Over the past 15 years, new nematicides that exhibit lower toxicity to vertebrates have been developed [8]. These include fluopyram, fluazaindolizine, fluensulfone, and cyclobutrifluram, which all have a trifluoro (3-F) group in the chemical structure [8]. Fluopyram was first developed as a fungicide and was shown later to be effective against PPNs. Fluopyram inhibits the function of succinate dehydrogenase (SDH), an enzyme that is essential to the mitochondrial respiratory chain [9,10]. Fluensulfone was shown to reduce root infection and plant penetration by PPN [11,12]. Although tested for their efficacy against PPN, some of the new nematicides such as fluazaindolizine require additional studies to characterize the mode of action [8,13]. Regarding cyclobutrifluram, the mode of action was already disclosed and classified in the IRAC (Insecticide Resistance Committee) and FRAC (Fungicide Resistance Committee) meeting in October 2022; however, a detailed study on its impact on development, reproduction, and transcriptional response is still lacking.

The well-established model organism, *Caenorhabditis elegans*, has become a very useful tool for screening for new drugs and studying their mode of action and their toxicity [9,14,15]. The majority of known anthelmintics are active against *C. elegans* [16,17]. Fluopyram was shown to increase oxidative stress, intestinal damage, and apoptosis in *C. elegans* [18]. Fluensulfone affects several aspects of *C. elegans* behavior [19].

In the present study, we determined the impact and the mode of action of cyclobutrifluram, hereafter named CB, using *C. elegans*. We investigated CB’s impact on the survival and fertility of *C. elegans*, and we found that the reproduction rate decreased as a result of low germ cell proliferation and increased the apoptosis level. Furthermore, two *C. elegans* strains that exhibit mutations in the C subunit of the SDH complex are resistant to CB treatment [9]. Based on these results, we propose that CB is targeting the mitochondrial SDH complex. Finally, using a transcriptomic approach, we showed that CB-treated worms present a strong transcriptional response, including the deregulation of genes coding for detoxifying enzymes.

## 2. Materials and Methods

### 2.1. C. elegans Strains and Culture

*C. elegans* wild-type strain (N2), VC294 *sdhb-1(gk165)/mIn1 [mIs14 dpy-10(e128)] II*, CU1546 *smIs34[ced-1p::ced-1::GFP+rol-6(su1006)]*, and *Escherichia coli* OP50 were provided by the Caenorhabditis Genetics Center (CGC). RP2698 *sdhc-1(tr407)* and RP2674 *sdhc-1(tr393)* were provided by Peter J. Roy (University of Toronto, Toronto, ON, Canada). All strains were grown on nematode growth medium (NGM) plates and fed with *E. coli* OP50 [20]. To obtain synchronized worms, eggs were isolated from gravid adults via bleaching [21].

### 2.2. Survival Assay

Cyclobutrifluram (88.9% active ingredient) was obtained from Syngenta Crop Protection (Basel, Switzerland). Stock solution of cyclobutrifluram was made in acetone (99.9%) and stored at 4 °C prior to use. Acetone was used as a control.

Lifespan assays were conducted using 100 age-synchronized worms at the L4 stage, which were transferred to new plates every day to avoid mixed generations. Various concentrations of cyclobutrifluram were applied. The surviving worms were counted daily. The experiment was conducted until the last worm was dead. Animals that bagged, exploded, or crawled off were censored. Statistical analysis was performed using the Kaplan–Meier survival assay (Oasis online application for survival analysis (Oasis)), and *p*-values were calculated using the log-rank test [22].

LC_50_ was calculated from a log-logistic regression curve with three parameters: the steepness of the dose–response curve, the upper asymptote, and the median lethal concentration (Quest Graph™ LC50 Calculator. AAT Bioquest, available at https://www.aatbio.com/tools/lc50-calculator (accessed on 14 September 2023)).

### 2.3. Brood Size and Embryonic Viability

Synchronized L4 hermaphrodite worms (N2, RP2698, and RP2674) were individually placed on NGM plates seeded with *E. coli* OP50 and exposed to various concentrations of cyclobutrifluram or fluopyram (Sigma-Aldrich, Burlington, MA, USA). The worms were transferred to new plates every 24 h until egg laying stopped. The total number of eggs, adult hermaphrodites, and males were scored. The average brood size, embryonic viability, and incidence of males were determined. Statistical significance was assessed using a two-tailed Student’s *t*-test with Welch’s correction, *p*-value ≤ 0.05.

### 2.4. Germline Apoptosis

To monitor apoptosis, synchronized L4 *smIs34[ced-1p::ced-1::GFP+rol-6(su1006)]* worms [23] exposed to 0.025 µM cyclobutrifluram for 24 h were observed using a widefield fluorescence microscope (Zeiss Axioplan 2, Oberkochen, Baden-Württemberg, Germany). The number of apoptotic corpses per gonad arm was scored as in [24].

### 2.5. Immunostaining and DAPI Staining

Synchronized L4 wild-type worms were exposed to 0.025 µM cyclobutrifluram and to acetone as a control. After 24 h, the worms were dissected to extrude the gonad, fixed, and immunostained following the protocol described in [25]. Anti-phospho-Histone H3 (Ser10), clone 3H10 from Sigma Aldrich was used as the primary antibody, and goat anti-mouse FITC from Jackson Immuno was used as the secondary antibody. The chromatin was stained with DAPI (4,6-diamidino-2-phenylindole hydrochloride). Scoring of mitotic cells was performed on 10–15 gonads for each condition.

### 2.6. Microscopy and Image Processing

Worms were examined under a widefield fluorescence microscope (Zeiss Axioplan 2 microscope, Zeiss AxioCam Color camera, AxioVision 4.6 software) and a confocal microscope (Leica TCS SP5 DM6000CS). The images were processed using Fiji ImageJ; the background was subtracted, and contrast/brightness was adjusted. Orientation of the images and final figure appearance was achieved using Adobe Photoshop (2023 v. 24.1.1) and Adobe Illustrator (2023 v. 27.1.1) software.

### 2.7. RNA Extraction, cDNA Library Preparation and Sequencing

Control and worms exposed to cyclobutrifluram (24 h) were collected as young adult hermaphrodites 50 h post-larval hatching after hypochlorite treatment (at 20 °C). The total RNA was extracted with TRIzol Reagent (Invitrogen; Carlsbad, CA, USA), and the RNA was purified using the PureLink RNA Mini Kit (Invitrogen; Carlsbad, CA, USA) according to the manufacturer’s instructions. cDNA library preparation and RNA sequencing were performed at the Next Generation Sequencing (NGS) Platform in Bern. The quality and concentration of each RNA sample and the following cDNA libraries were determined with a Qubit 2.0 fluorometer and the Fragment Analyzer CE12 AATI. RNA quality number (RQN) was higher than 8.5 for every sample. cDNA libraries were built using the TruSeq stranded mRNA library preparation kit (Illumina Inc.; San Diego, CA, USA). RNA sequencing (50 bp paired-end reads) was performed on two biological replicates per sample with the Illumina NovaSeq 6000 Sequencing System; cDNA libraries were multiplexed and sequenced in two lanes.

### 2.8. RNA-seq Data Analysis

The sequencing data were obtained from the Bern NGS platform. Raw reads in *.fasta* format were then uploaded to the Galaxy web platform using the public server at https://usegalaxy.org (accessed on 2 February, 2023) [26]. The quality of the reads was confirmed with FastQC, retrieved from http://www.bioinformatics.babraham.ac.uk/projects/fastqc/ (accessed on 2 February, 2023), and the reads were aligned to the *C. elegans* reference genome (version WS220) with TopHat (version 2.1.1)/Bowtie2 (version 2.2.8) [27,28] to obtain the *.bam* files. Read count by gene was obtained by HTSeq-count (version 0.9) [29], and differential gene expression analyses of the exposed group vs. control group (and thereby the principal component analysis and sample-to-sample distance plots) were performed using DESeq2 package [30]. Read counts were normalized by estimating size factors, and differential expression was tested against the negative binomial distribution using the Wald test. Multiple test correction was performed via an optimized false discovery rate (FDR) approach to obtain an adjusted *p*-value (*q* value) [31]. Genes were defined as differential-expressed genes (DEGs) with a *q* value ≤ 0.001 (minimum FDR) and −2 ≥ log_2_ fold change ≥ 2 cutoffs. The final list of significant DEGs was converted to an Excel sheet. The processed list, together with DESeq2 raw data outcome, are available in Appendix A.

Tissue and gene ontology (GO) term enrichment analysis for DEGs was performed using the Wormbase tool Gene Set Enrichment Analysis [32]. Enriched terms were found significant with an adj. *p*-value ≤ 0.05 obtained from the FDR correction using the Benjamini–Hochberg algorithm; significant GO terms (adj. *p*-value ≤ 0.05, Bonferroni correction) are shown. In addition, GO term functional analysis was performed using the database for annotation, visualization, and integrated discovery (DAVID version 6.8) NIAID/NIH tool [33]. Significant DAVID-GO terms (adj. *p*-value ≤ 0.05, Bonferroni correction) are shown in Appendix A.

Before uploading the DEGs to the Gene Set Enrichment Analysis and DAVID resources, gene names were updated to the last Wormbase release at the moment (WS283) using SimpleMine and Gene Name Sanitizer tools, which are available at https://wormbase.org/tools (accessed on 16 February 2023).

## 3. Results

### 3.1. Decreased Survival Rate of Worms Exposed to Cyclobutrifluram

To assess the effect of the nematicide cyclobutrifluram (Figure 1), we measured the median lethal concentration (LC_50_) for *C. elegans* worms 24 h after CB exposition and the lifespan of L4 worms exposed to three different concentrations of the compound (0.025, 0.1 and 0.25 µM). The LC_50_ was 0.069 µM (0.026 mg/L), which is about 10 times less than the LC_50_ of fluopyram for *C. elegans* (0.29 mg/L) [18]. All three concentrations of CB (0.025, 0.1, and 0.25 µM) significantly decrease the lifespan of the nematodes (Figure 2), although 0.025 µM of CB has a milder effect compared to the two other concentrations (0.1 µM and 0.25 µM). The mean lifespan of worms exposed to CB at the three concentrations 0.025, 0.1, and 0.25 µM) is 10, 1, and 2 days, respectively, vs. 16.5 days for the control worms. Altogether, these results showed that worms exposed to CB have a decreased lifespan.

### 3.2. Impact of Cyclobutrifluram on Brood Size and Embryo Viability

To assess the impact of CB on reproduction, we measured the brood size and embryo viability of worms exposed to three different concentrations of CB (0.025, 0.1, and 0.25 µM). Control worms lay, on average, 247 ± 70 eggs, and 99% of them give rise to viable embryos (Figure 3). Animals exposed to 0.025 µM of CB show a reduced number of progenies. However, the resulting embryos are 95% viable. Interestingly, their reproduction rate appears to be delayed compared to control worms. Low fertility is observed on the first three days of reproduction, followed by a higher rate of egg laying that ended after 6 days. Mild CB exposure (0.025 µM) appears to prolong the reproduction period of *C. elegans*, an observation that could be interpreted as a coping effect of the chemical. However, animals exposed to higher concentrations of CB (0.1 µM and 0.25 µM) are completely sterile (Figure 3). Altogether, the results show that CB has an impact on the reproduction abilities of *C. elegans* that depends on the concentration of the chemical in the medium.

### 3.3. Cyclobutrifluram Treatment Leads to a Decreased Number of Germ Cells

Worms treated with 0.025 µM of CB lay a limited number of eggs, suggesting that a high number of germline cells might be eliminated by apoptosis and/or germ cell proliferation might be impaired. At first, we inspected DAPI-stained gonads, which reveal, based on chromatin organization, the number and the progression of germline nuclei via the different stages of meiotic prophase, including the pairing of the homologous chromosomes in the transition zone (TZ) and the pachytene stage (PS) (Figure 4A). Overall, gonads of worms exposed to 0.025 µM CB show a normal progression via meiotic prophase but present a lower number of germ cells (Figure 4A). To monitor the number of apoptotic cells in the germline, we took advantage of a strain expressing fluorescent CED-1 protein, which is present at the membrane of the engulfing cells [23]. Control *ced-1::gfp* worms exhibit wild-type levels of apoptosis, while we observe a significant increase in the number of apoptotic germ cells in animals exposed to CB (Figure 4B,D). This result is consistent with the reduced number of germ cells that is observed in CB-treated worms.

The low number of germ cells could also arise from a decreased proliferation of germ precursor cells. To monitor the rate of germ cell proliferation, we used anti-phospho-serine10-histone 3 antibody (PH3) to highlight mitotic germ nuclei. By scoring PH3 positive nuclei, we were able to show that there is a decreased level of germ cell proliferation in worms exposed to 0.025 µM CB (Figure 4C,E). Altogether, these results show that CB has an impact on germ cell proliferation and the rate of development, as well as on germ cell survival.

### 3.4. Cyclobutrifluram Inhibits C. elegans Mitochondrial Complex II

Based on the similarity of CB chemical structure with fluopyram, another widely used nematicide, CB could potentially function as an inhibitor of the mitochondrial electron transport chain complex II [8]. To test this hypothesis, we took advantage of two *C. elegans* strains known to be resistant to fluopyram exposure [9]. These two strains, *sdhc-1(tr407)* and *sdhc-1(tr393)*, exhibit missense mutations in the gene coding for the C subunit of the succinate dehydrogenase enzymatic complex, also known as the mitochondrial electron transport chain complex II [9]. Wild-type worms, as well as *sdhc-1* mutants, were exposed to 0.25 µM of CB and fluopyram. Wild-type worms treated with CB and fluopyram show no progeny and a strongly reduced brood size, respectively (Figure 5B). Furthermore, about half of the embryos exposed to fluopyram are viable (Figure 5B). At the same concentration, CB exhibits a stronger effect on *C. elegans* reproduction rate compared to fluopyram. *sdhc-1(tr407)* mutants are partially resistant to both nematicides as they lay a higher number of eggs compared to wild-type worms exposed to both nematicides (Figure 5A,B). However, *sdhc-1(tr407)* mutants are still significantly impacted. On the other hand, exposed *sdhc-1(tr393)* mutants lay a similar number of embryos as *sdhc-1(tr393)* worms on control plates, indicating that the *sdhc-1(tr393)* mutant strain is fully resistant to both nematicide treatments (Figure 5A,B). Altogether, these results point towards the mitochondrial complex II as a target of CB.

To further test this idea, we compared the development of worms exposed to 0.025 µM with worms bearing mutations in the B subunit of the SDH complex. Both CB-treated wild-type worms and *sdhb-1* mutants exhibit a very similar developmental arrest at the L1/L2 stage with only very few germ precursor cells (Figure 5C). This observation is consistent with the idea that CB targets the SDH complex and inhibits its function.

### 3.5. Transcriptional Response of C. elegans Treated with Cyclobutrifluram

In order to investigate how CB is affecting the transcriptome of *C. elegans*, we isolated total mRNA from exposed and control worms. We found a total of 851 differentially expressed genes (DEGs) (*q* value ≤ 0.001 and −2 ≥ log_2_ FC ≥ 2). Out of these 851 genes, 570 are up-regulated, and 281 are down-regulated (Figure 6A). Gene ontology analysis using nematode-specific tools reveals that genes involved in metabolic processes and defense responses are over-represented in our dataset (Figure 6B) [32]. Interestingly, genes encoding cytochrome P450 (iron ion binding proteins) are enriched in the DEGs (Figure 6B and Appendix A). Cytochrome P450s are at the forefront of xenobiotics transformation. They contribute to their elimination from living organisms [34]. This step is followed by the action of UDP-glucuronosyl transferases (UGTs) and glutathione S-transferases (GSTs). Ten UGT genes were deregulated, one was up-regulated, and nine were down-regulated. We also performed a functional analysis using the DAVID tools, which is consistent with a strong metabolic response (Appendix A) [33,35]. Altogether, these results show that CB-treated worms respond by strongly deregulating genes encoding detoxifying proteins.

## 4. Discussion

Using *C. elegans* as a model organism, we show here that the new nematicide cyclobutrifluram impacts survival and reproduction abilities. Using various CB concentrations, we could demonstrate that the lifespan of the worms is decreased, and the number of progenies is reduced or absent. These results show a decreased germ cell proliferation rate and an increased level of apoptosis. Furthermore, we show that mutant worms exhibiting missense mutations in the C subunit of the SDH enzymatic complex are resistant to CB-treated worms. These results allow us to propose that CB targets the SDH complex of the mitochondrial respiratory chain. Finally, our transcriptomic analysis revealed that the worms present a strong transcriptional response to CB exposure.

Previous studies demonstrated that *Caenorhabditis elegans* is a useful model organism to test the impact of various drugs and also to determine their mode of action using the power of molecular genetic techniques [9,16]. Although cyclobutrifluram and its mode of action have been presented at several conferences, a detailed study regarding its impact on development, reproduction, and transcriptional response is missing [36,37,38]. Missense mutations in the C subunit of SDH could counteract the effect of CB on reproduction and development, indicating that CB acts by inhibiting the function of the SDH complex. Fluopyram, which exhibits a similar chemical structure to CB, also causes a decrease in the number of progeny in wild-type worms that is partially restored in *sdhc* mutants [18]. These nematicide-resistant strains were generated in the lab by mutagenizing the worms [9]. However, these mutations could potentially also arise in PPNs living in treated soil. Depending on the half-life of the nematicides, these could put selective pressure on the targeted nematodes and potentially induce mutations, rendering PPNs resistant to nematicide treatment. Additional studies will be required to determine the impact of CB on PPNs in the soil.

While assessing the impact of CB on the brood size of the worms, we noticed, in addition to the decreased brood size, a strong delay in the egg-laying rate. Wild-type *C. elegans* lay eggs during five days at 20 °C with a peak at day 2. However, worms exposed to 0.025 µM CB lay eggs for 6 days with a peak at day 5. One possible interpretation is that exposed worms are coping with the presence of CB in their growth medium.

Our transcriptomic analysis revealed a strong metabolic response to CB exposure. Enrichment analysis revealed that genes encoding cytochrome P450s and UGTs are highly represented (Appendix A). These detoxifying proteins add functional groups onto xenobiotic agents to solubilize them in the cell, facilitating their elimination. The over-representation of these enzymes in our dataset is consistent with a strong transcriptional response of the worm to CB exposure. However, we observed downregulation of 9 out of the 10 UGTs, which suggests that CB treatment induces only the expression of one specific UGT. Similarly, only a subset of UGTs were shown to be induced in *Meloidogyne incognita* upon treatment by nematicides such as fluensulfone and oxamyl [39]. Interestingly, genes encoding cytochrome P450s were also shown to be deregulated in *Meloidogyne incognita* upon treatment with various nematicides, including fluopyram [39]. *cyp-25A1* and *cyp-13A12* were found to be deregulated in PPN as in *C. elegans* upon exposure to various nematicides [39]. This shows that *C. elegans* shares similarities with PPN in its transcriptional response to nematicides and that it can be used to further investigate the impact of CB on nematodes.

Overall, our findings not only revealed the impact of CB on the reproduction and development of *C. elegans* but also the previously unknown molecular action of this nematicide that was not known so far. This study will contribute to further evaluating the potential resistance of nematodes against nematicides and to plan the elaboration of new drugs against PPNs.

## Figures and Tables

**Figure 1 jdb-11-00039-f001:**
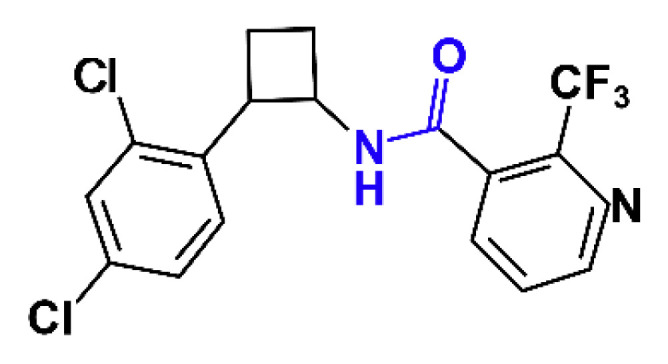
Chemical structure of cyclobutrifluram.

**Figure 2 jdb-11-00039-f002:**
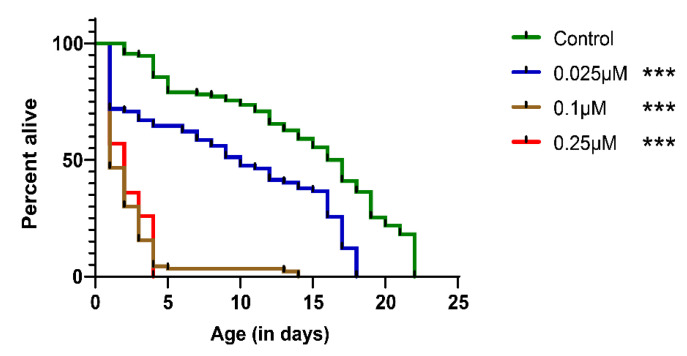
CB exposure decreases the lifespan of *C. elegans*. L4 larvae exposed to CB concentrations of 0.025 μM, 0.1 μM, and 0.25 μM show significant (*** *p*-value ≤ 0.001) decreased lifespan compared to control worms. Statistical comparisons were performed using the log-rank test.

**Figure 3 jdb-11-00039-f003:**
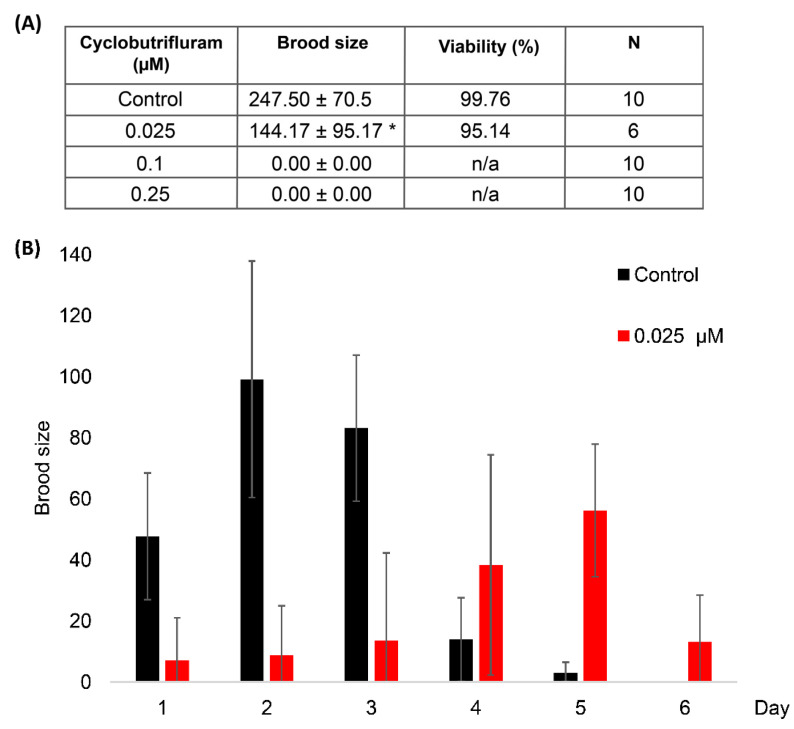
CB impacts *C. elegans* reproduction. (**A**) Table presenting the brood size and the percentage of embryo viability of worms exposed to different concentrations of CB. (**B**) Mean brood size of worms exposed to 0.025 µM CB and control worms plotted for each day. The asterisk in panel (**A**) indicates a significant difference from control worms. * *p*-value ≤ 0.05 (two-tailed Student’s *t*-test with Welch’s correction). n/a means not applicable.

**Figure 4 jdb-11-00039-f004:**
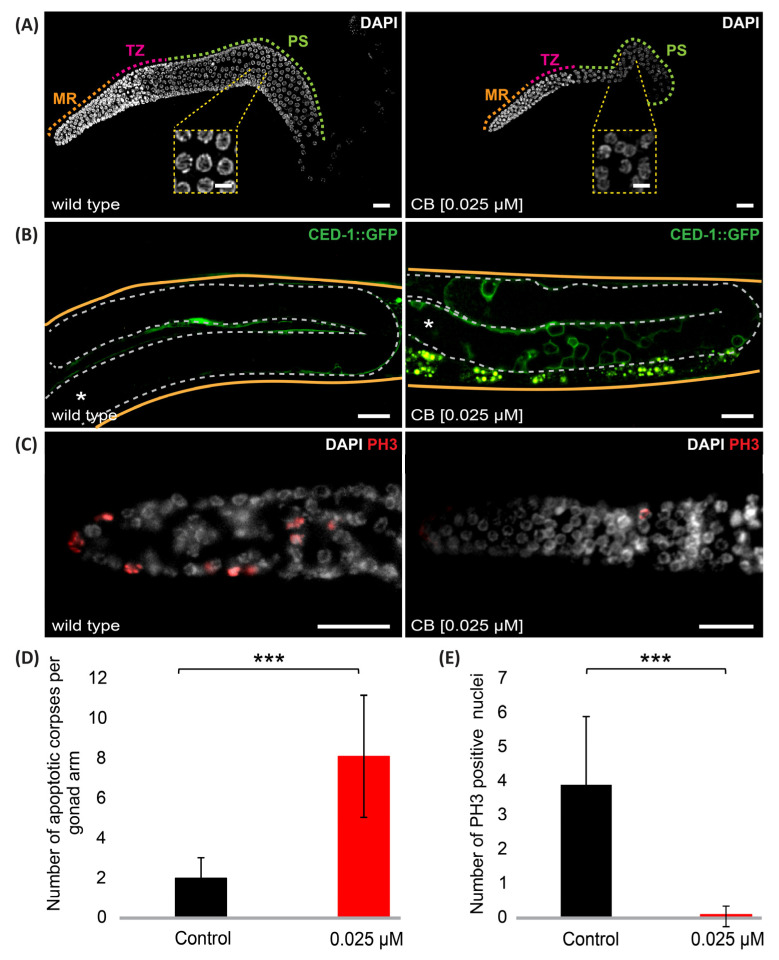
CB exposure leads to a decreased number of viable germ cells. (**A**) Representative confocal projection images of DAPI-stained gonads from control and CB (0.025 µM) exposed worms. (**B**) Control and CB (0.025 μM) exposed worms expressing CED-1::GFP. (**C**) Gonads from control and CB (0.025 μM) exposed worms immunostained with PH-3 antibody and stained with DAPI. (**D**) Histogram showing the number of apoptotic corpses per gonad arm for control and CB-exposed worms. (**E**) Histogram showing the number of PH3-positive nuclei in the mitotic region per gonad arm for control and CB-exposed worms. MR: mitotic region, TZ: transition zone, PS: pachytene stage. Data are presented as mean ± SD. *** *p*-value ≤ 0.001 (two-tailed Student’s *t*-test with Welch’s correction). Fifteen gonads were scored for each treatment. Scale bars, 20 µm and 5 µm for whole images and magnification panels, respectively. * Shows the distal part of the gonad.

**Figure 5 jdb-11-00039-f005:**
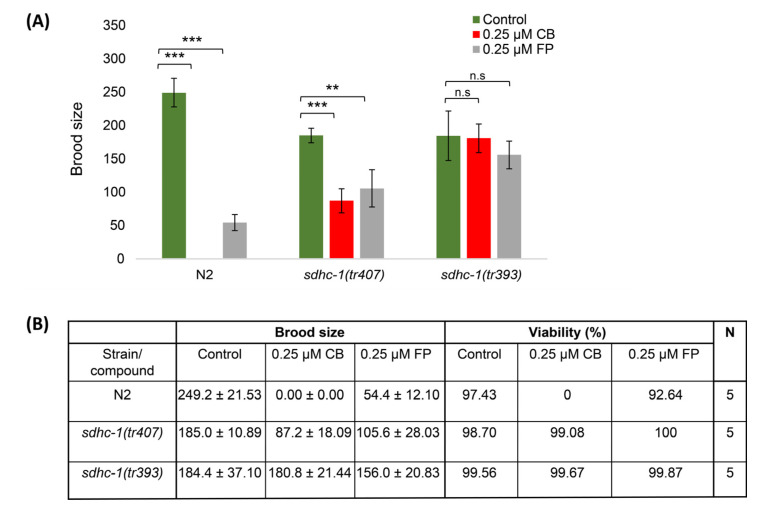
CB inhibits mitochondrial complex II. (**A**) Histogram showing the brood size of wild-type worms and *sdhc-1(tr407)* and *sdhc-1(tr393)* mutant strains exposed to 0.25 µM of CB and fluopyram (FP). (**B**) Table showing the brood size and the embryonic viability of wild-type worms and *sdhc-1(tr407)* and *sdhc-1(tr393)* mutant strains exposed to 0.25 µM of CB and FP. (**C**) Representative normarski pictures of a wild-type developmental arrested larva exposed to 0.025 µM CB and a *sdhb-1* mutant larva. Arrowheads point at germ nuclei. Data are presented as mean ± SD. *** *p*-value ≤ 0.001, ** *p*-value ≤ 0.01, ^n.s^ *p*-value > 0.05 (two-tailed Student’s *t*-test with Welch’s correction). Scale bars, 20 µm.

**Figure 6 jdb-11-00039-f006:**
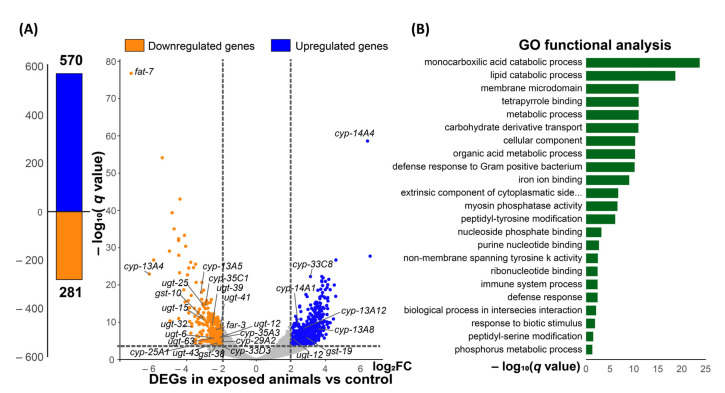
CB exposure leads to gene deregulation. (**A**) Bar and volcano plots show the number of significant deregulated genes in CB exposed vs. control worms determined by RNA-seq analysis. Dash lines indicate the significance and fold change cutoffs (*q* value ≤ 0.001 and −2 ≥ log_2_ fold change ≥ 2). (**B**) Graph illustrating the Gene Ontology term enrichment using T.E.A.-wormbase tool [32].

## Data Availability

Raw sequencing files of the RNA-seq experiment have been deposited in the ArrayExpress database at EMBL-EBI (www.ebi.ac.uk/arrayexpress (accessed on 12 July 2023)) under accession number E-MTAB-13189.

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
