# Peer review of "The New Nematicide Cyclobutrifluram Targets the Mitochondrial Succinate Dehydrogenase Complex in Caenorhabditis elegans"

_jdb, 2023, doi:10.3390/jdb11040039_

Round 1

Reviewer 1 Report

Understanding the mechanism of action of compounds that will likely be dispursed to the environment is important. Here, Heydari et al, present data supporting the idea that the mechanism of action of a relatively novel nematicide cyclobutrifluram is via inhibition of succinate dehydrogenase (SDH) in the model system C. elegans.

I would have liked to see the authors 'nail' the MOA by measuring SDH activity from mitochondrial preps +/- CB similar to Burns et al 2015 in a wt and sdhc mutant backgrounds to provide definitive evidence within the published literature. As it is, their strongest evidence supporting this MOA is that the fluopyram-resistant sdh mutants also resist CB. I would strongly encourage the authors to do this so that their paper will be the definitive paper that folks in the field will site when referring to CB MOA.

Similarly, I would have liked to have seen data summarized as EC50s, which is an expression that one can easily compare between compounds and nematodes.

I question whether any meaningful insight was provided with the analysis of the transcriptional response to CB beyond pure description.  There is a transcriptional response, yes, and CYPs are upregulated as expected. This is nice to know/report, but it is not particularly interesting on the face of it. Perhaps repeating the analysis with Fluopyram might yield insight- what is simply a xenobiotic response vs what might be the transcriptional response to inhibition of SDH.  If I had to pick one area of improvement for the manuscript though, it would be measuring SDH activity +/-CB. 

Minor:

Figure 3- error bars should be shown.

Figure 5a- '0,25' should be changed to '0.25'

Reviewer 2 Report

This manuscript describes the effects of treatment of C. elegans with cyclobutrifluran (CB), a nematicide that is being developed to target plant parasitic nematodes. The authors found that the CB treatment shortens lifespan of C. elegans and interferes with C. elegans germ cell development, leading to decreased germ cell proliferation and increased germ cell apoptosis. A C. elegans succinate dehydrogenase mutant (sdhc-1) is resistant to the effects of CB. This work is important, because it helps identify the targets of this new class of nematicide, and it suggests that nematode reproductive processes are sensitive to this nematicide.

A few issues need to be addressed:

1.       Since untreated schd-1 mutants show reduced brood size, it would be interesting to do the DAPI staining, CEP-1 assays, and proliferation assays on these mutants to determine if genetic down-regulation of succinate dehydrogenase affects germline develop in similar ways to the CB treatment.

2.       In Figure 5C, a picture of a WT worm treated with 0.025uM CB is shown and described as L1/L2 arrest.  However, all of the other assays (lifespan, germline characterizations) are used in adult WT worms treated with 0.025uM CB.  Thus, apparently a portion of these treated worms arrest before reaching adulthood, and only those worms reaching adulthood are used for the lifespan and germline assays. I think it is important to report the percentage of arrested larvae with treatment of each concentration of CB.

A few grammatical errors, for example, in the Abstract line 12:, it should be “specifically target”.

Reviewer 3 Report

The manuscript investigates the impact of the nematicide cyclobutrifluram on the nematode Caenorhabditis elegans. Plant-parasitic nematodes (PPNs) pose a significant threat to agriculture and food security. Traditional nematicides have had broad effects on various organisms, including non-target species, making them environmentally hazardous. In contrast, newer nematicides like cyclobutrifluram have shown specificity towards PPNs. The study aims to understand the mode of action of cyclobutrifluram using C. elegans as a model organism and explore its potential as a novel nematicide.

The main findings are: 1/ Cyclobutrifluram exposure decreases the lifespan and fertility of C. elegans. Higher concentrations of cyclobutrifluram result in complete sterility, while lower concentrations cause a delay in reproduction. 2/ Cyclobutrifluram treatment leads to a reduced number of germ cells due to increased apoptosis and decreased germ cell proliferation. 3/ The authors proposes that cyclobutrifluram functions by inhibiting the mitochondrial succinate dehydrogenase (SDH) complex, similar to the nematicide fluopyram. 4/ Transcriptomic analysis reveals a strong response to cyclobutrifluram exposure in C. elegans, including deregulation of genes coding for detoxifying proteins, such as cytochrome P450s and UDP-glucuronosyl transferases (UGTs).

The results highlight the potential of cyclobutrifluram as a promising nematicide with a specific mode of action against PPNs. The study also confirms the utility of C. elegans as a suitable model organism to study the mode of action of nematicides.

Major concerns:

The impact of CB on reproduction was clearly demonstrated, but it should have been accompanied by DIC analysis of the gonad to examine overall gonad organisation/architecture. Additionally, ovulation rates should be analysed. The authors raise the valid point that CB could start to get degraded after three days, and this may result in the improve reproductive output. It would be simple to test this hypothesis by transferring he worms to fresh CB plates.

The authors claim that DNA damage by Cyclobutrifluram results in increase apoptosis. However, the authors have not directly examined if the increase rate of apoptosis is via the physiological or damage-induced pathways. The authors must use common ced and egl-1 mutants to determine which apoptotic pathway is the cause of the increased germ cell death.

The images of Figure 4 are not high enough. Consider having increase resolution/blow ups of specific regions to demonstrate the chromosomal organization.

Given the hypothesis similar mode of action (inhibitor of the mitochondrial electron transport chain complex II), the RNA-seq analyses would have been more powerful if fluopyram was included alongside Cyclobutrifluram.

Minor points:

Below is a list of some but not all formatting errors that need to be fixed.

P 73: C. elegans is not italicized.

P73. C. elegans gene formatting nomenclature is not followed.

P74. Escherichia coli is not italicized.

P77. E. coli is not italicized.

P 90. p-value.

P94. Escherichia coli written in full.

P293. The RNA-seq analysis is interesting. Why would 9 UGTs be down regulated by exposure to CB? Would you not expect up regulation of these genes to help xenobiotics transformation? Are there examples of down regulation of UGTs upon exposure to xenobiotics?

The English is good- only a few minor issues to fix.

Round 2

Reviewer 1 Report

The authors did not address any of my concerns experimentally.

Author Response

I am aware that we did not complete the manuscript with the experimental measurement of SDH activity. It would have been an additional evidence that CB is targeting SDH. However, we do not have the resources to perform these experiments.

Reviewer 2 Report

I am satisfied with the revisions made by the authors.

Author Response

Thanks for the reply. I am glad that we could address all the comments.